# Risks and Benefits of Prophylactic Transfusion before Cholecystectomy in Sickle Cell Disease

**DOI:** 10.3390/jcm11143986

**Published:** 2022-07-09

**Authors:** Elise Rambaud, Brigitte Ranque, Sofia Tsiakyroudi, Laure Joseph, Nathalie Bouly, Richard Douard, Anne François, Jacques Pouchot, Jean-Benoît Arlet

**Affiliations:** 1Reference Center for Sickle Cell Disease, Thalassemia and Other Red Blood Cell and Erythropoiesis Diseases, Department of Internal Medicine, European Georges Pompidou University Hospital (AP-HP), European Georges Pompidou Hospital (AP-HP), Paris Cité University, F-75015 Paris, France; brigitte.ranque@aphp.fr (B.R.); jacques.pouchot@aphp.fr (J.P.); jean-benoit.arlet@aphp.fr (J.-B.A.); 2Digestive Surgery Department, European Georges Pompidou University Hospital (AP-HP), F-75015 Paris, France; sofia.tsiakyroudi@aphp.fr (S.T.); richard.douard@aphp.fr (R.D.); 3Reference Center for Sickle Cell Disease, Thalassemia and Other Red Blood Cell and Erythropoiesis Diseases, Biotherapy Service, Necker Hospital (AP-HP), F-75015 Paris, France; laure.joseph@aphp.fr; 4French Blood Establishment, F-75015 Paris, France; nathalie.bouly@efs.sante.fr (N.B.); anne.francois@efs.sante.fr (A.F.)

**Keywords:** sickle cell disease, transfusion, cholecystectomy, vaso-occlusive crisis, acute chest syndrome

## Abstract

Preoperative transfusion (PT) reduces acute postoperative vaso-occlusive events (VOE) in sickle cell disease (SCD), but exposes patients to alloimmunization, encouraging a recent trend towards transfusion sparing. The aim of this study was to investigate the benefit–risk ratio of PT before cholecystectomy on the occurrence of postoperative VOE. Adult SCD patients who underwent cholecystectomy between 2008 and 2019 in our center were included. Patients’ characteristics, collected retrospectively, were compared according to PT. A total of 79 patients were included, 66% of whom received PT. Gallbladder histopathology found chronic cholecystitis (97%) and gallstones (66%). Transfused patients underwent more urgent surgeries and had experienced more painful vaso-occlusive crises (VOC) in the month before surgery (*p* = 0.05). Four (8.5%) post-transfusion alloimmunizations occurred, and two of them caused a delayed hemolytic transfusion reaction (DHTR) (4.3%). The occurrence of postoperative VOE was similar between the groups (19.2% vs. 29.6%, *p* = 0.45). Though not statistically significant, a history of hospitalized VOC within 6 months prior to surgery seemed to be associated to postoperative VOE among non-transfused patients (75% vs. 31.6%, *p* = 0.10). PT before cholecystectomy exposes to risks of alloimmunization and DHTR that could be avoided in some patients. Recent VOCs appear to be associated with a higher risk of postoperative VOE and prompt the preemptive transfusion of these patients.

## 1. Introduction

Sickle cell disease (SCD) leads to chronic hemolysis, which is responsible for the accumulation of unconjugated bilirubin, which may result in cholelithiasis. The prevalence of pigmentary gallstones increases with age, reaching 50% in adulthood. They cause acute biliary complications in 10 to 20% of cases [1,2,3,4,5]. These complications, in addition to their own severity, also increase the incidence of vaso-occlusive events (VOE): painful vaso-occlusive crises (VOC) and acute chest syndromes (ACS) [6]. In the United Kingdom and the United States, cholecystectomy is recommended for SCD adults in case of symptomatic cholelithiasis [7,8]. In France, a cholecystectomy is recommended for all SCD patients (adults and children) with a demonstrated ultrasound cholelithiasis, even if asymptomatic, before the occurrence of any complication [9,10,11]. However, this surgery is at risk in this specific population since global stress, lack of oxygenation, and acidosis related to the surgical procedure may trigger VOC or ACS. The latter is a serious complication, with mortality ranging from 1.6 to 4.6% in hospitalized patients, according to previous studies [9,12,13,14]. Abdominal surgery is particularly associated with an increased risk of postoperative ACS, reaching up to 30% in non-transfused patients [13,14,15]. Peri-operative management therefore aims at avoiding these VOE as much as possible. Prophylactic preoperative transfusion (PT), which can be either a simple transfusion or exchange transfusion depending on the hemoglobin (Hb) level, reduces these postoperative VOE in adult SCD patients [13,16,17,18]. However, it exposes patients to alloimmunization and delayed hemolytic transfusion reactions (DHTR), the latter being a potentially lethal complication which explains a recent trend towards transfusion sparing [7,19,20]. However, whether PT can be avoided without risk for a simple surgery such as cholecystectomy is uncertain [21].

The primary objective of our study was to investigate the benefit–risk ratio of PT before a cholecystectomy on the occurrence of postoperative complications in a cohort of adult SCD patients, and to determine the factors associated with an increased risk of postoperative vaso-occlusive complications. The secondary objective was to describe patient management during a cholecystectomy.

## 2. Materials and Methods

We conducted a single-center, retrospective, and observational study in the adult Referral Center for SCD of the European Georges Pompidou University Hospital in Paris (France). Our hospital’s computerized data warehouse allowed for an exhaustive search of all consecutive adult SCD patients who underwent a cholecystectomy in our hospital between 1 January 2008 and 31 December 2019 (defined by codes D57.0, D57.01, D57.02, D57.09 for SCD and K82.9, Z90.4, Y83.6 for cholecystectomy, Table A1).

For all patients followed in the referral center, socio-demographic data, background treatments, acute and chronic complications of the SCD, comorbidities, and routine laboratory tests at steady state were prospectively collected. Nephropathy was defined as chronic renal failure (glomerular filtration rate < 60 mL/min/1.73 m^2^) or a urine albumin to creatinine ratio (ACR) > 3 mg/mmol on at least two measures at steady state, more than 6 months apart. Retinopathy was defined as any retinal change secondary to retinal ischemia and included proliferative and non-proliferative forms. Surgical and pathological data were collected from operative and pathological reports. Preoperative VOE were defined as all painful bone VOC and/or ACS requiring hospitalization within 6 months before surgery. Patients who underwent surgery in an emergency setting were differentiated from those who underwent elective planned surgery. VOE in the immediate postoperative period and up to one year after the surgery, the need for admission in the intensive care unit (ICU) and orotracheal intubation, as well as deaths and the need for re-hospitalization in the month following the surgery were retrospectively collected.

Concerning the transfusion data, the French Blood Establishment database was used to confirm the dates of red blood cells (RBC) transfusion, the number of RBC units received, the emergence of post-transfusion allo-antibodies, and their type. Any new allo-antibody that had an impact on the choice of the type of RBC during subsequent transfusions, or triggered a delayed hemolytic transfusion reaction (DHTR) (defined as the appearance, at least 3 days after a RBC transfusion, of biological markers of hemolysis associated with a deeper anemia than before transfusion (minimum decrease 30%), and a significant drop in hemoglobin A percentage) was considered significant.

The primary endpoint was the occurrence of significant VOE (painful VOC and/or ACS requiring an extension of hospital stay or readmission) or death in the month following surgery. Secondary endpoints included transfusion-related complications (alloimmunization, DHTR), postoperative complications other than VOE, need for readmission, and need for postoperative transfusion. The parameters studied were compared between two groups of patients (preoperative transfusion or not, postoperative complication or not) by Student or Wilcoxon test for quantitative variables according to their distribution, and by Chi-2 or Fischer test for qualitative variables, according to the numbers. The non-transfused patients who did and did not present postoperative complications were then compared. We also performed the same analyses in the sub-population of patients with the S/C genotype. A multivariate analysis by logistic regression was performed to search for factors associated with the occurrence of a postoperative complication in the whole sample. Given the small number of events, the models tested included only two explanatory variables: preoperative transfusion and another variable (Hb phenotype (S/S and S/β^0^ vs. other), VOC in the previous month, ACS in the previous 6 months or lifetime, and acute cholecystitis or not). *p* values less than 0.05 were considered significant. All analyses were performed with the R software (Ross Ihaka, Robert Gentleman, Auckland, New Zealand) (version 4.0.3, 2020).

The local institutional board approved this study on 26 March 2020 (IRB registration number #00011928, Comité d’Ethique de la Recherche Assistance Publique Hôpitaux de Paris (CERAPHP.5)) as non-interventional research using data collected for routine clinical practice. In line with the French legislation on retrospective studies of routine clinical practice, participants were not required to give their written informed consent. Patients included in this study were all informed that their medical data could be used for research purposes, in accordance with General Data Protection Regulation 2016/679.

## 3. Results

### 3.1. Patients’ Characteristics and Transfusion Rate

A total of 79 SCD adult patients (43% male) were included. The median age was 27.6 years (IQR 23.6–32.5) and 64 (81%) were of the S/S or S/β^0^-thalassemia genotype. The main characteristics of the patients are reported in Table 1. All patients underwent a laparoscopic cholecystectomy, with a median total length of hospital stay of 4 days (IQR 3–9.5). The pathological analysis of the gallbladder found lesions of chronic cholecystitis (*n* = 73, 97%) and the presence of gallstones (*n* = 50, 66%). Lesions of acute or subacute cholecystitis were observed in 15 patients (20%).

The trend over the study period of the percentage of patients with SCD who received pre-cholecystectomy PT is reported in Figure 1. Although the decrease is statistically not significant (*p* = 0.20), we observed that PT in our center is no longer systematic since 2010. There was no other major management practice change during the study period.

### 3.2. Comparison According to Prophylactic Transfusion

We compared the 52 patients (66%) who received PT before a cholecystectomy with those who did not (Table 1). Transfused patients were more likely to have a S/S or S/β^0^ genotype (88.5% vs. 66.7%, *p* = 0.04) and to receive hydroxyurea therapy (42.6% vs. 23.8%, *p* = 0.13). There was no other difference in demographics, chronic complications, or history of ACS before surgery between transfused and non-transfused patients. The pre-procedure hemoglobin levels (pre-operative or pre-PT) were not statistically different between the two groups (9.2 vs. 9.5 g/dL, *p* = 0.59). However, transfused patients had more preoperative vaso-occlusive complications, including more VOC in the month before surgery (39.2% vs. 14.8%, *p* = 0.05), with the same trend in the 6 preoperative months (62.7% vs. 44.4%, *p* = 0.19). They were more often operated in an emergency setting (38.5% vs. 14.8%, *p* = 0.06). The median length of stay was comparable between the two groups: 4 (3.8–10) days in the transfused group versus 3 (3–6.5) days in the non-transfused group (*p* = 0.11).

The prevalence of VOE (VOC or ACS or death) in the postoperative month between transfused and non-transfused patients were not statistically different (19.2% vs. 29.6%, *p* = 0.45) (Table 2). One non-transfused S/C patient died of an unexplained coma one day after surgery. Four patients in the transfused group (7.7%) required readmission within one month of discharge: one for VOC, one for VOC and DHTR, one for DHTR, and one for a postoperative collection requiring revision surgery. In the non-transfused group, five patients (18.5%) were readmitted (*p* = 0.26 compared to transfused patients): four for ACS and one for VOC. Five patients required postoperative transfusion, four (7.7%) in the transfused group and one (3.7%) in the non-transfused group (*p* = 0.84).

Four patients acquired new allo-antibodies with implications for subsequent transfusions (8.5%) in the transfused group: anti RH4, RH6, MNS1, MNS3 (*n* = 1), anti JKB (*n* = 1), anti RH10 private (*n* = 1), and anti RH2 KEL3 and MNS3 (*n* = 1). Two of these patients (4.3% of transfused patients), 34- and 23-year-old S/S patients, developed DHTR at 11 and 19 days of prophylactic transfusion, respectively, and both required readmission. The lowest hemoglobin levels for these two patients were 5 g/dL and 6.7 g/dL, respectively. Considering all severe complications (hospitalized VOC or ACS or death or alloimmunization) in the postoperative month, the incidence was not statistically different between the two groups (26.9% in transfused vs. 29.6% in non-transfused).

### 3.3. Comparison of Non-Transfused Patients

We then focused on the 27 patients who did not receive PT. As already shown, 8 of them (29.6%) developed VOC (*n* = 1), ACS (*n* = 6), or died (*n* = 1). Only one of these patients was transfused postoperatively, without secondary alloimmunization, for an ACS that occurred at day 8 after surgery, without secondary alloimmunization. We compared the characteristics of these 8 patients with the 19 non-transfused patients who did not develop any postoperative complications. The sociodemographic characteristics, proportion of S/S or S/β^0^-thalassemia patients, prevalence of chronic complications, and history of ACS were comparable between the 2 groups (Table 3). Non-transfused patients with a history of VOC hospitalized within 6 months before surgery were more frequent among the non-transfused patients who developed post-operative complications: 6/8 (75%) versus 6/19 (31.6%) among those who did not (*p* = 0.10). The median time from last VOE to surgery was 105 (52.3–204) days in patients with postoperative complications, versus 270 (120–366) days in other non-transfused patients (*p* = 0.14). Patients operated on in an emergency setting also seemed to undergo more VOE (25% vs. 10.5%), although this was not statistically significant (*p* = 0.71).

A total of 34 patients of our cohort had neither hospitalized VOC within 6 months preoperatively nor emergency surgery. These patients had fewer postoperative VOE, regardless of their preoperative transfusion status: 13.3% (2/15) in non-transfused patients and 15.8% (3/19) in transfused patients. Among these 19 patients with PT, there were 2 (10.5%) transfusion-related complications: 1 alloimmunization and 1 DHTR.

### 3.4. Characteristics of Patients Experiencing VOE

Regardless of PT status, 18 patients (22.8%) had experienced a postoperative VOE within one month after surgery (*n* = 17) or death (*n* = 1). The characteristics of these patients are shown in Table A2. Overall, 13 (72.2%) had a history of VOCs within 6 months before the surgery versus 30/59 (50.8%) in those without post-operative VOE (*p* = 0.18). Moreover, 8 (44.4%) were operated in an emergency setting versus 15 (25%) in patients without post operative VOE, (*p* = 0.20). A gallbladder histopathological analysis revealed more frequent lesions of acute or subacute cholecystitis in patients with post-operative VOE (7/18, 38.9% vs. 8/57, 14%, *p* = 0.05).

### 3.5. Outcome in S/C Population

Our population included 11 patients with the S/C genotype (63% male) and a median age of 35.8 years (IQR 30.2–39.9). None was receiving hydroxyurea therapy at the time of the surgery. They had a comparable history of ACS as patients with S/S and S/β^0^ genotypes (8/11, 72.7%, vs. 44/64, 68.8%, *p* = 1). PT was performed in 5 (45.5%) of them versus 46 (71.9%) in patients with S/S and S/β^0^ genotypes (*p* = 0.15). Two patients with the S/C genotype (18%), who had not been transfused, experienced severe postoperative VOE: an ACS at D3 postoperative that required readmission (without the need of a transfusion), and a death at day 1 postoperative because of an unexplained coma, without associated VOE.

## 4. Discussion

This study shows an evolution of practice in our referral center towards transfusion sparing since 2010. We did not find any statistically significant difference in terms of prevalence of occurrence of VOC or ACS or death in the postoperative month between patients who did or did not receive PT for a cholecystectomy. In the randomized trial published in 2013 by Howard et al. [16] comparing prophylactic transfusion before various low and medium-risk surgeries (including 44% abdominal surgeries) in 70 patients, there were 39% postoperative complications in non-transfused patients compared with 15% complications in transfused patients postoperatively. The complication rate in non-transfused patients was higher than in our study (39% vs. 29.6%): this can be explained by several factors. First, the retrospective nature of our study induces a bias of indication of PT, with an enrichment of less severe SCD phenotypes in the group of patients without PT even when considering patients with S/S and S/β^0^-genotype, as in the study by Howard et al. (27.7% of complication rate). Second, in our study, the proportion of patients treated by hydroxyurea therapy was higher (36.8% vs. 11%).

In our study, the most frequent VOE that occurred after surgery were ACS (44.4% of all VOE). This rate is comparable to that found in the study by Howard et al. and is consistent with numerous studies finding an increased risk of ACS perioperatively in abdominal surgery, although almost all of the surgeries were performed by laparoscopy [13,14,21]. One possible pathophysiological explanation is the use of carbon dioxide to perform the pneumoperitoneum, which may cause postoperative hypercapnia [15].

Our study also shows that the practice of transfusion sparing has prevented transfusion complications. The rates of alloimmunization and DHTR in our series (8.5% and 4.3%, respectively) among the transfused patients are consistent with the results of the literature (7.5% of alloimmunization in the study by Howard et al., but no DHTR). If VOC or ACS can be managed (including with postoperative transfusion, if necessary), alloimmunization, in addition to the immediate lethal risk of DHTR (5 to 10% depending on the series) [22,23,24], often may condemn the possibilities of subsequent transfusions in these young patients during severe VOE.

The analysis of the 8/27 non-transfused patients with postoperative complications (29.6%) revealed that these patients did not have more preoperative history of ACS than patients without postoperative complications (75% vs. 73.7%) but had more frequently been hospitalized for a VOC in the past 6 months (75% vs. 31.6%). The latter result was not statistically significant, possibly because of a lack of power due to the small sample size. However, it is reasonable to assume that a recently hospitalized crisis may be a ground for postoperative vaso-occlusive complications.

The occurrence of a postoperative VOE in our series was more frequent in patients operated in an urgent setting (44%). In a large American pediatric study evaluating the risk factors associated with the occurrence of postoperative vaso-occlusive complications in abdominal surgery, the urgent nature of the surgery was the only significant predictive factor of complications (OR 1.83, 95% CI 1.02–3.29, *p* = 0.04), whether the patients were transfused or not [25]. Likewise, in the French series conducted at Tenon Hospital by Muroni et al. (103 patients), postoperative vaso-occlusive complications were less frequent in the case of a prophylactic laparoscopic cholecystectomy in asymptomatic patients (11.5%) than in patients with an acute vesicular complication (25.5%) [26]. Significant differences were also found by Currò et al. (study that included 30 patients) in terms of operative time, morbidity rate, postoperative length of stay, and total length of hospitalization between children operated before the onset of abdominal symptoms and children operated after the onset of symptoms [27]. These patients are therefore at greater risk of postoperative complications and could clearly benefit from PT.

The modalities of perioperative management have rarely been evaluated in patients with the S/C genotype: two trials specifically excluded these patients [16,28], and one trial included only nine of them [20]. However, the risk of vaso-occlusive complications seems to be increased in case of abdominal surgery in these patients [29]. Our study, though it included a limited number of S/C patients (*n* = 11), shows that the rate of postoperative complications is significant (18% of patients, including one unexplained death). This indicates that S/C patients are also at risk of complications in the peri-operative period and could also benefit from prophylactic transfusion in case of recent VOE.

As mentioned above, the main limitation of our study is related to the indication bias for PT that artificially reduces the difference of post-operative complications between transfused and non-transfused patients. In addition, the data were recorded retrospectively, but the existence of a prospective database in our center systematically collecting SCD data at each appointment in our center allowed us to minimize the risk of missing data. Nevertheless, the reasons for indication (or non-indication) of transfusion were unfortunately not specified in the files. Furthermore, the small size of the study induces a lack of statistical power, although the sample size is comparable to the main studies studying the benefit of preoperative prophylactic transfusion in patients with SCD [16,30].

## 5. Conclusions

Transfusion sparing before a planned cholecystectomy seems to be a safe option in properly selected stable SCD patients, with no history of hospitalization for VOC within the 6 previous months, and in the context of planned surgery. This targeted strategy could allow avoiding potentially dangerous alloimmunizations for the long-term management of sickle cell patients. It should therefore be tested in a randomized controlled study, taking both the short-term risk of VOE and the long-term risk of alloimmunization into account.

## Figures and Tables

**Figure 1 jcm-11-03986-f001:**
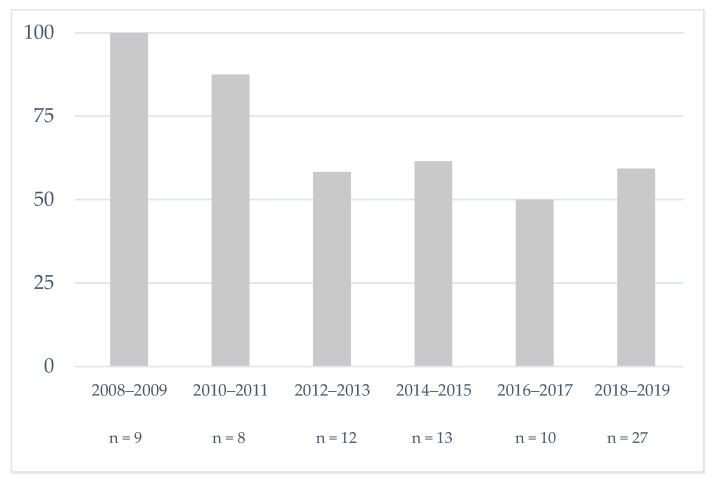
Evolution of the percentage of sickle cell patients who received pre-cholecystectomy prophylactic transfusion in our center.

**Table 1 jcm-11-03986-t001:** Patients’ characteristics and comparison according to prophylactic transfusion.

	Overall *n* = 79	PT *n* = 52	No PT *n* = 27	*p*
Age	27.6 (23.6–32.5)	27.1 (23.7–31.5)	28.6 (23.6–34.7)	0.57
Sex (M)	34/79 (43.0)	24/52 (46.2)	10/27 (37)	0.60
BMI (kg/m^2^)	21.8 (19.7–24.2)	21.6 (19.4–24.5)	22.3 (20.3–24.1)	0.58
**Sickle cell genotype**				
S/S + S/β^0^-thalassemia	64/79 (81.0)	46/52 (88.5)	18/27 (66.7)	0.04
S/β^+^-thalassemia	4/79 (5.1)	1/52 (1.9)	3/27 (11.1)	0.11
S/C	11/79 (14.0)	5/52 (9.6)	6/27 (22.2)	0.17
Current hydroxyurea treatment among S/S and S/β^0^ patients ^1^	25/68 (36.8)	20/47 (42.6)	5/21 (23.8)	0.13
Dose (mg) (mean ± SD)	1110 ± 399.2	1175 ± 398.2	1000 ± 223.6	0.10
Preoperative alloimmunization	6/73 (8.2)	3/51 (5.9)	3/22 (13.6)	0.36
Pre-procedure Hb level (g/dL) ^2^	9.2 (7.9–10.2)	9.2 (7.8–9.9)	9.5 (8.8–11.4)	0.59
Preoperative transfusion	52/79 (66.0)	-	-	-
Number of transfused RBC units	2 (2–2)	2 (2–2)	-	-
Urgent surgery	24/79 (30.4)	20/52 (38.5)	4/27 (14.8)	0.06
**Preoperative acute VOE**				
VOC in the past 1 month	24/79 (30.4)	20/51 (39.2)	4/27 (14.8)	0.05
VOC in the past 6 months	44/79 (55.7)	32/51 (62.7)	12/27 (44.4)	0.19
ACS in the past 6 months	18/79 (22.8)	13/51 (25.5)	5/27 (18.5)	0.68
VOC and/or ACS in the past 6 months	45/79 (56.9)	32/51 (62.7)	13/27 (48.1)	0.32
Time between the last VOE and surgery (days)	142.5 (30–366)	90.0 (30–366)	240.0 (75–366)	0.13
History of ACS, lifetime	55/79 (69.6)	35/52 (67.3)	20/27 (74.1)	0.72
Number of previous ACS	1 (0–2)	1 (0–3)	1 (0.5–1.5)	0.45
**Chronic complications with percentage > 20%**				
Nephropathy	16/79 (20.3)	12/52 (23.1)	4/27 (14.8)	0.55
Retinopathy	22/79 (27.8)	15/52 (28.8)	7/27 (25.9)	0.78
Osteonecrosis	19/79 (24.1)	11/52 (21.2)	8/27 (29.6)	0.4
Priapism (in men only)	8/34 (23.5)	6/24 (25.0)	2/10 (20.0)	1
Length of hospital stay (days)	4 (3–9.5)	4 (3.75–10)	3 (3–6.5)	0.11

Values are expressed as median (IQR) or n/N (%), unless specified. ACS: acute chest syndrome; BMI: body mass index; RBC: red blood cells; Hb: hemoglobin; PT: prophylactic transfusion; VOC: vaso-occlusive bone crisis; VOE: vaso-occlusive event. ^1^ No current hydroxyurea treatment among patients with S/C genotype. ^2^ Pre-PT for transfused patients, pre-operative for non-transfused patients.

**Table 2 jcm-11-03986-t002:** Postoperative complications according to prophylactic transfusion.

	Overall *n* = 79	PT *n* = 52	No PT *n* = 27	*p*
**Postoperative complications**				
VOC and/or ACS and/or death	18/79 (22.8)	10/52 (19.2)	8/27 (29.6)	0.45
ICU admission	5/79 (6.3)	3/52 (5.8)	2/27 (7.4)	1
Postoperative infection (except ACS)	6/79 (7.6)	5/52 (9.6)	1/27 (3.7)	0.66
**Transfusion-related complications**				
New antibodies	4/68 (5.9)	4/47 (8.5)	0/21 (0)	0.3
DHTR with antibodies	2/68 (2.9)	2/47 (4.3)	0/21 (0)	1
Postoperative transfusion less than 1 month after surgery	5/79 (6.3)	4/52 (7.7)	1/27 (3.7)	0.84
Number of RBC units	2 (2–3)	2.5 (1.75–3.25)	2	-
Readmission less than 1 month after surgery	9/79 (11.4)	4/52 (7.7) ^1^	5/27 (18.5) ^2^	0.26
Time between surgery and readmission (days)	9 (4–21.3)	10 (7–23)	8 (3–16)	0.86
Death	1/79 (1.3)	0	1/27 (3.7)	0.34

Values are expressed as median (IQR) or n/N (%) unless specified. ACS: acute chest syndrome; DHTR: delayed haemolytic transfusion reaction; ICU: intensive care unit; RBC: red blood cells; PT: prophylactic transfusion; VOC: vaso-occlusive bone crisis. ^1^ Reason for readmission: 1 VOC, 1 DHTR, 1 VOC and DHTR, 1 postoperative collection. ^2^ Reason for readmission: 4 ACS and 1 VOC.

**Table 3 jcm-11-03986-t003:** Characteristics of patients without prophylactic transfusion according to the occurrence of postoperative complications.

	No Complication *n* = 19	VOE or Death after Surgery *n* = 8	*p*
Age	28.5 (23–32.3)	29.6 (23.7–37.2)	0.22
Sex (M)	7/19 (36.8)	3/8 (37.5)	1
BMI (kg/m^2^)	21.5 (3.8)	22.8 (3.3)	0.65
**Sickle cell genotype**			
S/S + S/β^0^-thalassemia	13/19 (68.4)	5/8 (62.5)	1
S/β^+^-thalassemia	2/19 (10.6)	1/8 (12.5)	1
S/C	4/19 (21.0)	2/8 (25.0)	1
Current hydroxyurea treatment among S/S and S/β^0^ patients ^1^	4/13 (30.8)	1/6 (17.0)	1
Dose (mg) (mean ± SD)	930 ± 125	500	
Urgent surgery	2/19 (10.5)	2/8 (25.0)	0.71
**Preoperative acute VOE**			
VOC in the past 6 months	6/19 (31.6)	6/8 (75.0)	0.10
ACS in the past 6 months	3/19 (15.8)	2/8 (25.0)	0.98
Time between the last VOE and surgery (days)	270 (120–366)	105 (52.3–204)	0.14
History of ACS, lifetime	14/19 (73.7)	6/8 (75.0)	1
Number of previous ACS	1 (0.5–1.5)	1 (0.75–1.25)	0.68
Length of hospital stay (days)	3 (2.2)	10.5 (7.5)	<0.001

Values are expressed as median (IQR) or n/N (%) unless specified. ACS: acute chest syndrome, BMI: body mass index, VOC: vaso-occlusive bone crisis, VOE: vaso-occlusive event. ^1^ No current hydroxyurea treatment among patients with S/C genotype.

## Data Availability

The data presented in this study are available on request from the corresponding author.

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
