# Peer review of "Risks and Benefits of Prophylactic Transfusion before Cholecystectomy in Sickle Cell Disease"

_jcm, 2022, doi:10.3390/jcm11143986_

Round 1

Reviewer 1 Report

Overall the manuscript is well  written.  The only concern is author has to add more number of sample data for NoPT and also describe in the discussion section about why chose only male sex  in this study. 

Author Response

Response to Reviewer 1 Comments

Point 1 : Overall the manuscript is well written.  The only concern is author has to add more number of sample data for NoPT and also describe in the discussion section about why chose only male sex in this study. 

Response : Concerning data for no PT patients, we included all patients who underwent cholecystectomy during the study period. The inclusion was exhaustive thanks to the hospital's computerized data warehouse (as already stated in the Method section of the manuscript). It is therefore not possible to add more number of sample data for these patients in the study period.

We did not choose to include specifically male sex patients in the study. When studying the overall population of the study (n = 79), there were 34 male patients (43%). This data is stated in table 1. There was no statistically significant difference in sex between transfused and non transfused patients (p = 0.6).

Reviewer 2 Report

The authors present a very detailed analysis of the association of different factors and the outcome of cholecystectomy according whether prophylactic transfusion was administered or not.
The conclusion drawn is that a transfusion sparing approach is justifiable as it avoids alloimmunisation while the complication rate between the 2 groups is not statistically significant.
However, from their detailed analysis there is a salient factor missing: The pre-procedure haemoglobin level. In clinical practice the degree/severity of anaemia is a major factor in decision-making regarding transfusion before surgery.
We cannot conclude that there is no significant difference between post-operative complications unless we know the degree of anaemia of patients and how that influenced the decision to transfuse preoperatively.

Author Response

Response to Reviewer 2 Comments

Point 1 : The authors present a very detailed analysis of the association of different factors and the outcome of cholecystectomy according whether prophylactic transfusion was administered or not.
The conclusion drawn is that a transfusion sparing approach is justifiable as it avoids alloimmunisation while the complication rate between the 2 groups is not statistically significant.

However, from their detailed analysis there is a salient factor missing: The pre-procedure haemoglobin level. In clinical practice the degree/severity of anaemia is a major factor in decision-making regarding transfusion before surgery.

We cannot conclude that there is no significant difference between post-operative complications unless we know the degree of anaemia of patients and how that influenced the decision to transfuse preoperatively.

Response : We thank the reviewer for this remark. We agree that pre-procedure hemoglobin level is an essential factor that influences the preoperative transfusion decisions.

We thus have collected this missing data (pre-operative hemoglobin levels for non-transfused patients, and pre-transfusion hemoglobin levels for transfused patients), and found that there was no statistically significant difference between transfused and non-transfused patients (median hemoglobin levels 9.2 vs 9.5 ; p = 0.59).

We have therefore included this information into table 1.

We have aldo added a sentence in paragraph 3.2 in order to precise this information « Pre-procedure hemoglobin levels (pre-operative or pre-PT) were not statistically different between the two groups (9.2 vs 9.5 g/dL, p = 0.59). »

Round 2

Reviewer 2 Report

Text adequately revised